# Non-Thermal Plasma Attenuates TNF-α-Induced Endothelial Inflammation via ROS Modulation and NF-κB Inhibition

**DOI:** 10.3390/ijms26094449

**Published:** 2025-05-07

**Authors:** Joo-Hak Kim, Seonhee Kim, Shuyu Piao, Minsoo Kim, Dae-Woong Kim, Byeong Hwa Jeon, Sang-Ha Oh, Cuk-Seong Kim

**Affiliations:** 1Department of Plastic and Reconstructive Surgery, Sejong Chungnam National University Hospital, Sejong 30099, Republic of Korea; joohak82@gmail.com; 2Department of Physiology & Medical Science, School of Medicine, Chungnam National University, Daejeon 34134, Republic of Korea; wlxlsunny@naver.com (S.K.); piaoshuyu@cnu.ac.kr (S.P.); g4qsol@naver.com (M.K.); bhjeon@cnu.ac.kr (B.H.J.); 3Department of Plasma Engineering, Korea Institute of Machinery and Materials (KIMM), Daejeon 34134, Republic of Korea; matinblue@gmail.com; 4Department of Plastic and Reconstructive Surgery, Chungnam National University Hospital, Daejeon 34134, Republic of Korea

**Keywords:** non-thermal plasma, endothelial cell inflammation, reactive oxygen species, NF-κB

## Abstract

Non-thermal plasma (NTP) has emerged as a promising therapeutic tool due to its anti-inflammatory properties; however, its molecular effects on vascular endothelial inflammation remain unclear. This study investigated the effects of NTP on tumor necrosis factor-alpha (TNF-α)-induced inflammation in human umbilical vein endothelial cells (HUVECs). NTP treatment significantly reduced intracellular reactive oxygen species (ROS) levels and downregulated the expression of adhesion molecules such as intercellular adhesion molecule-1 (ICAM-1) and vascular cell adhesion molecule-1 (VCAM-1), which are key markers of endothelial activation. In addition, NTP suppressed mRNA expression of pro-inflammatory cytokines, including TNF-α, interleukin-1 beta (IL-1β), and interleukin-6 (IL-6). Mechanistically, NTP inhibited the nuclear translocation of phosphorylated NF-κB p65, indicating attenuation of NF-κB signaling. These results demonstrate that NTP modulates inflammatory responses in endothelial cells by attenuating ROS generation and suppressing NF-κB-mediated signaling. Our findings suggest that NTP may serve as a potential therapeutic strategy for treating vascular inflammation and related pathologies.

## 1. Introduction

Plasma medicine has emerged as an interdisciplinary frontier, captivating the attention of researchers from engineering, physics, biochemistry, and biotechnology [1,2,3,4]. Among the various forms of plasma, non-thermal plasma (NTP) has gained significant interest due to its unique characteristics and diverse therapeutic potentials. NTP is an ionized gas generated at or near room temperature, operating under atmospheric pressure without inducing substantial thermal damage to the surrounding environment [5]. These properties render NTP particularly suitable for direct biological applications, including wound healing, cancer therapy, bacterial sterilization, and blood coagulation [6,7,8,9]. Unlike thermal plasma, which poses risks of tissue damage due to heat, NTP enables safe and localized treatment, making it a modality in medicine.

Inflammation, a hallmark of the immune response, plays a critical role in acute and chronic disease states. Inflammation is triggered by infection, injury, or cellular stress, leading to the release of cytokines and other signaling molecules. While inflammation is a critical component of the immune response, excessive or chronic inflammation can lead to the development of chronic diseases, including cancer, cardiovascular disorders, and neurodegenerative conditions [10,11,12]. In cardiovascular disease, endothelial cells are key regulator of vascular homeostasis and inflammation [13,14,15]. The activation of endothelial cells expresses adhesion molecules such as intercellular adhesion molecule-1 (ICAM-1) and vascular cell adhesion molecule-1 (VCAM-1), inducing leukocyte adhesion [16,17]. In addition, activated endothelial cells produce pro-inflammatory cytokines including factor-alpha (TNF-α), interleukin (IL)-1β, and IL-6, perpetuating the inflammatory cycle [18].

The NF-κB signaling pathway is a key regulator of the inflammatory response, modulating the expression of genes involved in cytokine production, leukocyte recruitment, and cell survival [19,20,21]. Dysregulation of the pathway is linked to chronic inflammatory diseases, highlighting its importance as a therapeutic target. Although NTP has been shown to exhibit anti-inflammatory effects in various models, including skin inflammation and lung injury, its specific effects on endothelial inflammation [22,23] and the precise intracellular mechanisms involved remain poorly understood. In particular, the role of NTP in modulating NF-κB signaling and the downstream targets in endothelial cells have not been thoroughly investigated.

This study aims to elucidate the anti-inflammatory effects of NTP on human umbilical vein endothelial cells (HUVECs) stimulated with TNF-α. We hypothesized that NTP might suppress endothelial inflammation by inhibiting key components of the NF-κB signaling cascade, including the nuclear translocation of phosphorylated p65 (P-p65). Our results demonstrate that NTP treatment reduces the expression of pro-inflammatory cytokines and adhesion molecules while attenuating NF-κB signaling. These findings suggest that NTP may serve as a novel therapeutic strategy for managing vascular inflammation and related diseases.

## 2. Results

### 2.1. Characterization of Non-Thermal Plasma (NTP)

A schematic of the plasma generation system and a photograph of the NTP device are shown in Figure 1A,B. The NTP was generated using a custom-designed nozzle connected to a plasma production module capable of controlling gas input (e.g., air, helium, nitrogen) under atmospheric pressure. The plasma consists primarily of reactive oxygen and nitrogen species. Optical emission spectra (OES) were recorded using a high-resolution spectrometer (HR4000, Ocean Optics, Dunedin, FL, USA) with a 150 ms integration time. The fiber-optic input port of the spectrometer was positioned 2 mm from the plasma tip to ensure accurate and reproducible readings. As shown in Figure 1C,D, key emission peaks included nitric oxide (245 nm), nitrogen (315 nm), and oxygen (845 nm), confirming the generation of reactive nitrogen and oxygen species.

To further examine the behavior of reactive species over time, temporal changes in emission intensity were evaluated. Figure 1E shows a gradual increase in NO emission intensity over time, while nitrogen and oxygen emissions also demonstrated consistent profiles (Figure 1F,G). These data confirm the system’s stability and capability to produce biologically relevant reactive species during plasma generation, ensuring consistent exposure throughout experimentation.

### 2.2. NTP Treatment Does Not Compromise HUVEC Viability or Proliferation

To investigate the safety profile of NTP exposure, HUVECs were exposed to NTP for 0, 3, 10, or 30 s. Following a 12 h incubation period, changes in cell morphology were observed via phase-contrast microscopy (Figure 2A). The treated cells retained their characteristic cobblestone morphology, and no evidence of detachment or cellular shrinkage was observed across treatment groups.

Quantitative evaluation of viability (Figure 2B) and proliferation (Figure 2C) showed no significant differences between control and NTP-treated groups, regardless of exposure duration. These findings were consistent across three independent experiments. These results confirm the biocompatibility of NTP treatment in endothelial cells.

### 2.3. NTP Treatment Inhibits TNF-α-Induced ROS Generation and Inflammatory Cytokine Expression by HUVECs

To determine the anti-inflammatory potential of NTP, HUVECs were pre-exposed to NTP for varying durations (0, 3, 10, or 30 s), followed by stimulation with TNF-α (10 ng/mL) for 60 min and subsequent 12 h incubation. TNF-α alone significantly elevated ROS levels, while NTP pretreatment reduced this ROS generation in a time-dependent manner (Figure 3A). The strongest suppression was observed with 30 s NTP treatment.

Next, we assessed the expression of key pro-inflammatory cytokines. TNF-α stimulation increased mRNA levels of TNF-α, IL-1β, and IL-6, confirming a strong inflammatory response. NTP pretreatment significantly attenuated these effects. IL-1β expression was particularly sensitive, showing marked reduction even at 3 s exposure, while TNF-α and IL-6 expression required longer NTP exposure (10–30 s) for effective suppression (Figure 3B–D). These results suggest that NTP modulates TNF-α-induced inflammatory signaling by reducing oxidative stress and transcriptional upregulation of inflammatory cytokines.

### 2.4. NTP Treatment Inhibits TNF-α-Induced Expression of Inflammatory Cytokines and Monocyte Adhesion by HUVECs

TNF-α stimulation is known to promote endothelial activation by inducing expression of adhesion molecules such as VCAM-1 and ICAM-1. As shown in Figure 4A,B, TNF-α markedly elevated the mRNA expression of both molecules in HUVECs. NTP pretreatment resulted in a dose-dependent reduction in VCAM-1 and ICAM-1 transcripts. Western blot analysis confirmed that this transcriptional suppression was accompanied by a corresponding reduction in protein levels (Figure 4C). Densitometric quantification (Figure 4D,E) supported the observed protein-level changes.

To investigate the functional impact of these changes, U937 monocyte adhesion assays were performed. As expected, TNF-α stimulation significantly increased monocyte adhesion to HUVECs. However, NTP pretreatment reduced monocyte binding in a time-dependent manner (Figure 4F), with 30 s exposure nearly restoring adhesion levels to baseline (Figure 4G). These findings demonstrate that NTP not only downregulates adhesion molecule expression but also inhibits monocyte-endothelial interactions, a key step in vascular inflammation and immune cell recruitment.

### 2.5. NTP Treatment Inhibits TNF-α-Induced NF-κB-p65 Signaling by HUVECs

To explore the mechanistic basis of NTP’s anti-inflammatory effect, we examined the impact on the NF-κB signaling pathway. In control HUVECs, TNF-α stimulation induced phosphorylation of p65 and degradation of IκBα, resulting in the translocation of p65 into the nucleus (Figure 5A,B). Pretreatment with NTP preserved IκBα expression and reduced p65 phosphorylation, indicating inhibition of the canonical NF-κB activation cascade.

To confirm this, cytoplasmic and nuclear fractions were isolated to evaluate p65 distribution. TNF-α caused a marked shift of p65 from the cytoplasm to the nucleus. However, NTP pretreatment significantly inhibited this nuclear translocation (Figure 5C). Immunofluorescence imaging further confirmed these observations: in control cells, p65 was primarily cytoplasmic, while TNF-α caused nuclear accumulation. NTP prevented this translocation, maintaining cytoplasmic localization of p65 (Figure 5D). These results demonstrate that NTP suppresses NF-κB activation at multiple levels, including inhibition of upstream signaling (IκBα degradation), reduced phosphorylation of p65, and impaired nuclear translocation. These findings support the potential of NTP as a modulator of inflammatory transcriptional pathways in vascular endothelial cells.

## 3. Discussion

In this study, we extensively investigated the anti-inflammatory effects of non-thermal plasma (NTP) on vascular endothelial cells and provided mechanistic insights into the therapeutic potential for vascular inflammatory disorders. Our results demonstrate that NTP significantly attenuates endothelial inflammation through multiple biological pathways, including the modulation of reactive oxygen species (ROS), suppression of pro-inflammatory cytokines, downregulation of inflammatory signaling cascade such as the nuclear factor-kappa B (NF-κB) pathway (Figure 6).

A central observation in this study was the marked reduction in ROS levels in endothelial cells treated with TNF-α following NTP exposure (Figure 3A). ROS are essential mediators in various cellular processes, including signaling and homeostasis [24,25]. However, excessive ROS can lead to oxidative stress, contributing to endothelial dysfunction and the progression of vascular diseases such as atherosclerosis and hypertension. By the inhibition of ROS production, NTP restores cellular redox balance, thereby preventing the activation of oxidative stress-related inflammatory pathways. This reduction in ROS not only alleviates the inflammatory response but also protects endothelial cells from oxidative damage, promoting vascular function and structural integrity [26].

Although DCFDA-based ROS detection is utilized due to the sensitivity of DCFDA, it is prone to artifacts such as auto-oxidation and cell density-dependent effects. Therefore, the results of ROS quantification should be interpreted cautiously. Moreover, while our findings suggest a correlation between reduced ROS levels and suppression of NF-κB signaling, the causality between these two events has not been directly established in this study. Future investigations employing ROS scavengers, such as N-acetylcysteine (NAC), or ROS inducers, such as hydrogen peroxide (H_2_O_2_), will be essential to confirm the mechanistic link between ROS modulation and NF-κB inhibition.

In addition, our study demonstrated that NTP treatment significantly suppressed the expression of pro-inflammatory cytokines, including TNF-α, IL-1β, and IL-6 (Figure 3). These cytokines are central players in the inflammatory response, and their overproduction can lead to chronic inflammation, tissue damage and progression of vascular diseases [10]. The inhibition of these cytokines by NTP suggests a robust anti-inflammatory action, which we attributed to the modulation of the NF-κB signaling pathway. NF-κB, a pivotal transcription factor, regulates the expression of various genes involved in inflammation [20]. Our data showed that NTP inhibits the translocation of the NF-κB p65 subunit to the nucleus, thereby reducing the transcriptional activity of inflammation-related genes. This finding is consistent with the role of NF-κB as a regulator of immune responses and highlights the potential of NTP in modulating the NF-κB signaling pathway.

In addition to cytokine suppression, NTP treatment was shown to decrease the expression of adhesion molecules such as VCAM-1 and ICAM-1 (Figure 4). These adhesion molecules facilitate the adhesion and recruitment of leukocytes to the endothelium, a key step in the initiation and progression of vascular inflammation. NTP reduced monocyte adhesion to endothelial cells, as confirmed by monocyte adhesion assays in this study (Figure 4). The NTP effect on adhesion molecules not only alleviates vascular inflammation but also addresses one of the earliest events in the inflammatory cascade, potentially preventing the progression of endothelial dysfunction and associated diseases. In this study, the anti-inflammatory effects of NTP are consistent with previous studies that reported similar findings across various cell types and tissues. For instance, NTP has been shown to reduce inflammation in lung epithelial cells and skin tissues, indicating anti-inflammatory properties of NTP [8,27,28,29]. Importantly, our study also confirmed that NTP treatment does not compromise endothelial cell viability or proliferation, a critical consideration for clinical applications. The ability of NTP to exert anti-inflammatory effects while maintaining cell viability and proliferation makes it a particularly promising candidate for further preclinical and clinical development.

Although this study provides strong evidence, the exact molecular mechanisms behind the anti-inflammatory effects of NTP are still not fully understood. Future research should focus on identifying the specific reactive species generated by NTP and their individual contributions to modulating inflammatory signaling pathways. Additionally, studies exploring the interaction between NTP and other signaling cascades associated with inflammation will provide a more comprehensive understanding of the therapeutic potential of NTP. In vivo studies are also essential to validate the translational applicability of NTP in clinical settings and to optimize the NTP delivery for treating vascular inflammation and other inflammatory conditions.

In conclusion, our study demonstrates that NTP exerts non-cytotoxic anti-inflammatory effects in vascular endothelial cells by reducing ROS levels, suppressing pro-inflammatory cytokines and adhesion molecules, and inhibiting NF-κB signaling. These findings suggest that NTP may represent a promising potential therapeutic approach for mitigating endothelial inflammation.

However, several limitations should be considered. First, the exclusive use of HUVECs limits the generalizability of the results; future studies should validate the effects of NTP in primary endothelial cells and in vivo models. Second, although we focused on the NF-κB p65 subunit, we did not investigate other subunits such as p50 or related pathways like MAPK signaling. Third, while MTT and CCK-8 assays showed no cytotoxicity, sublethal cellular stress could not be fully excluded; thus, additional assays such as LDH release and mitochondrial membrane potential measurements are warranted.

Future studies addressing the points will be crucial to further elucidate the molecular mechanisms of NTP and to translate the therapeutic potential into clinical applications for vascular inflammatory diseases.

## 4. Materials and Methods

### 4.1. Cell Culture

Human umbilical vein endothelial cells (HUVECs) were purchased from Clonetics (San Diego, CA, USA) and cultured in endothelial growth medium-2 from Lonza (Walkersville, MD, USA) according to the manufacturer’s instructions at 37 °C with 5% CO_2_. Sub-confluent, proliferating HUVECs at passages 2–8 were used. U937 monocytes were purchased from Clonetics (San Diego, CA, USA) and cultured in RPMI 1640 media (Welgene, Gyeongsan-si, Republic of Korea) according to the manufacturer’s instructions at 37 °C with 5% CO_2_.

### 4.2. NTP Treatment

A single-nozzle, torch-type NTP equipment from PARA Korea (Seoul, Republic of Korea) was used. The distance between the nozzle of equipment and the bottom of cell culture plate was fixed at 30 mm. HUVECs were prepared in 6-well culture plates 24 h before NTP treatment.

The plasma was generated using ambient air as the working gas without additional carrier gases. Optical emission spectra were measured at two different power settings: P1 (low power) and P10 (high power). These settings were selected to compare plasma stability and reactive species production efficiency, with P10 showing stronger emission intensities for reactive nitrogen and oxygen species.

Cells were exposed directly to NTP for different time points (0 s, 3 s, 10 s and 30 s) at 5 different spots in each well and incubated for 30 min followed by stimulation with 10 ng/mL tumor necrosis factor-α (TNF-α) (Sigma Aldrich, St. Louis, MO, USA) and incubation for another 12 h at 37 °C. After 12 h incubation, subsequent experiments were performed.

### 4.3. Cell Viability and Cell Proliferation Assay

Cell viability and cell proliferation after time-dependent NTP treatment in HUVECs, was measured using MTT assay (3-(4, 5-dimethyl-2-thiazolyl)-2, 5-diphenyl-2H-tetrazolium bromide (Sigma Aldrich, St. Louis, MO, USA)) and CCK-8 assay (DOJINDO Molecular Technologies, Tokyo, Japan), respectively, according to the manufacturer’s instructions. The optical density of each well was measured using a microplate reader (TECAN, Männedorf, Switzerland) at 590 nm and 450 nm, respectively.

### 4.4. Western Blotting

An amount of 30 μg of whole-cell lysate was loaded and separated on 8–10% SDS-PAGE gels by electrophoresis, followed by incubation in the appropriate primary and secondary antibodies. For each Western blot quantified, the experiment was repeated for a minimum of three times. Blots were imaged using a chemiluminescence assay kit (Miracle-Star Western Blot Detection System; Intron Biotechnology, Seongnam, Republic of Korea) and EZ-Western Lumi Femto (Daeil Lab Service, Seoul, Republic of Korea), and band densities were quantified on a Gel Doc 2000 Chemi Doc system using Quantity One 1-D software (Bio-Rad, Hercules, CA, USA). Values were normalized to β-actin (loading control). The following primary antibodies were used: anti-VCAM-1, anti-ICAM-1, anti-iNOS, anti-NF-κB p65, anti-IκBα and anti-PARP were all purchased from Santa Cruz Biotechnology (Santa Cruz, CA, USA), anti-p-NF-κB p65 from Abcam (Cambridge, MA, USA) and β-actin from Cell Signaling Technology (Beverly, MA, USA).

### 4.5. Immunofluorescence

HUVECs were cultured on glass coverslips in 6-well plates, and NTP was treated time-dependently followed by stimulation with TNF-α and incubation for 12 h. After washing with phosphate-buffered saline (PBS), cells were fixed with 4% (*w*/*v*) paraformaldehyde for 20 min and then permeabilized with 0.3% (*v*/*v*) Triton X–100 for 30 min at room temperature. After blocking with PBS containing 5% (*w*/*v*) bovine serum albumin for 1 h, cells were incubated with NF-κB p65 antibody (1:50) overnight at 4 °C and then labeled with Fluorescein (FITC) secondary antibody (1:200) for 1 h in the dark at room temperature. Images were obtained using a fluorescence microscope (Leica, Wetzlar, Germany).

### 4.6. Preparation of Nuclear and Cytosolic Fractions

HUVECs were treated with NTP time-dependently, followed by stimulation with TNF-α and incubation for 12 h. Nuclear and cytosolic fractions were isolated using the NE-PER extraction kit (Pierce, Rockford, IL, USA) according to the manufacturer’s instructions. The cells were lysed and centrifuged at 12,000 rpm for 10 min. The supernatant (cytosolic fraction) and nuclear pellet were collected. The cytosolic fraction was lysed in 20 mM/L Tris buffer (pH 7.5) containing 0.5 mM/L EDTA-Na2, 0.5 mM/L EGTA-Na2, and protease inhibitors. The supernatant fractions were collected and used for Western blotting of p65-NF-κB. The nuclear pellet was re-suspended in nuclear buffer (20 mM HEPES, pH 7.9, 0.4 M NaCl, 1 mM EDTA, 10% glycerol, 1 mM DTT, and protease inhibitors), vortexed, and centrifuged at 12,000 rpm for 20 min. The supernatant fractions were collected and used for Western blotting of p65-NF-κB and IκBα.

### 4.7. Monocyte Adhesion Assay

After discarding the culture medium, HUVECs were incubated with a suspension of non-activated U937 monocytes in HBSS at a monocyte-to-endothelial cell ratio of approximately 5:1. The incubation was performed for 45 min at 37 °C under static conditions. After incubation, non-adherent monocytes were removed by washing 2–3 times with HBSS, and the number of adherent monocytes was quantified by manual counting.

### 4.8. Real-Time Polymerase Chain Reaction (qPCR)

Total RNA from HUVECs was isolated using TRIzol Reagent (Invitrogen, Carlsbad, CA, USA, USA) according to the manufacturer’s instructions. The RNA concentration was quantified using a SmartSpec3000 spectrophotometer (Bio-Rad, Hercules, CA, USA). To detect mRNA expression levels, RNA was reverse-transcribed into cDNA synthesis by using 5x cDNA Master Mix (CellSafe, Yongin, Republic of Korea). qPCR was performed using the Prism 7000 Sequence Detection System (Applied Biosystems, Faster City, MA, USA) with SYBR Green One-Step qPCR SuperMix (Enzynomics, Daejeon, Republic of Korea). qPCR conditions were set as follows: 95 °C for 90 s followed by 40 cycles of 95 °C for 15 s, 60 °C for 20 s and 72 °C for 30 s. All primers were purchased from COSMO GENETECH (Seoul, Republic of Korea). The primer sequences are shown in Table 1.

### 4.9. Detection of Total ROS

HUVECs were treated with NTP time-dependently, followed by stimulation with TNF-α and incubation for 12 h. After trypsinization and neutralization, the suspended cells were incubated with 10 μM DCFDA for 30 min at 37 °C. Next, cells were washed with PBS, and the fluorescent signal was measured by Fluoroskan Ascent fluorescence reader (Thermo Fisher Scientific, Waltham, MA, USA) at 475 nm excitation and 500~550 nm emission.

### 4.10. Data Analysis

Statistical analysis was performed using Prism 8 software (GraphPad Software, San Diego, CA, USA). Data are presented as the mean ± standard deviation. Differences between two groups were evaluated using unpaired *t*-tests. *p* < 0.05 were considered to indicate a statistically significant difference. Data are representative of at least three independent experiments.

## Figures and Tables

**Figure 1 ijms-26-04449-f001:**
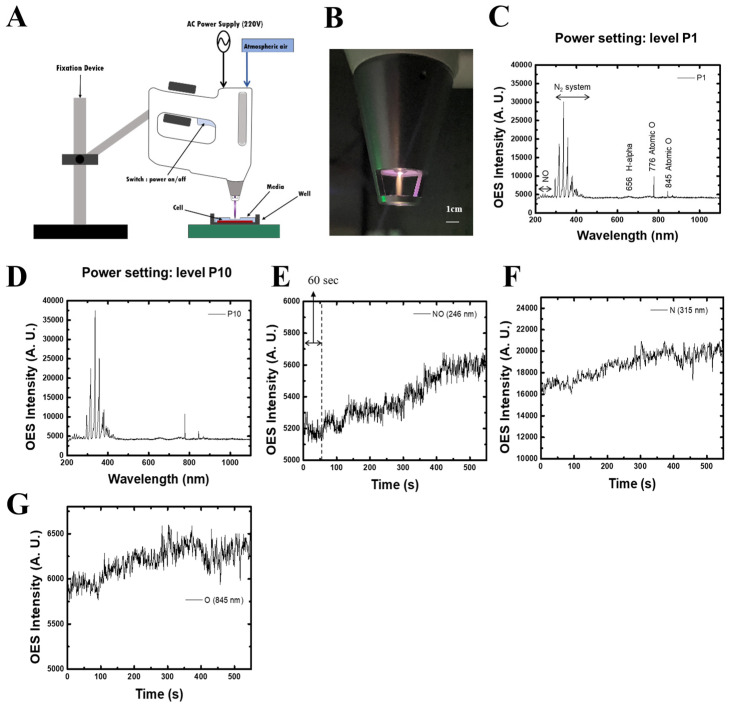
Schematic representation and characterization of the NTP device. (**A**) Schematic diagram illustrating the non-thermal plasma (NTP) generation system, including the optical emission spectrometer (OES) setup and the atmospheric air plasma jet system. The system operates under atmospheric conditions to produce NTP for experimental applications. (**B**) Photograph of the NTP device generating plasma. (**C**) Optical emission spectrum (OES) intensity measured at a power setting of P1. (**D**) OES intensity at a higher power setting (P10). (**E**–**G**) Temporal dynamics of reactive species generated during plasma discharge. (**E**) Nitric oxide (NO) emission intensity at 246 nm over time. (*X*-axis: time [seconds], *Y*-axis: emission intensity [arbitrary units]). (**F**) Nitrogen (N) emission at 315 nm and (**G**) atomic oxygen (O) emission at 845 nm showing time-dependent emission intensity (*X*-axis: time [seconds], *Y*-axis: emission intensity [arbitrary units]).

**Figure 2 ijms-26-04449-f002:**
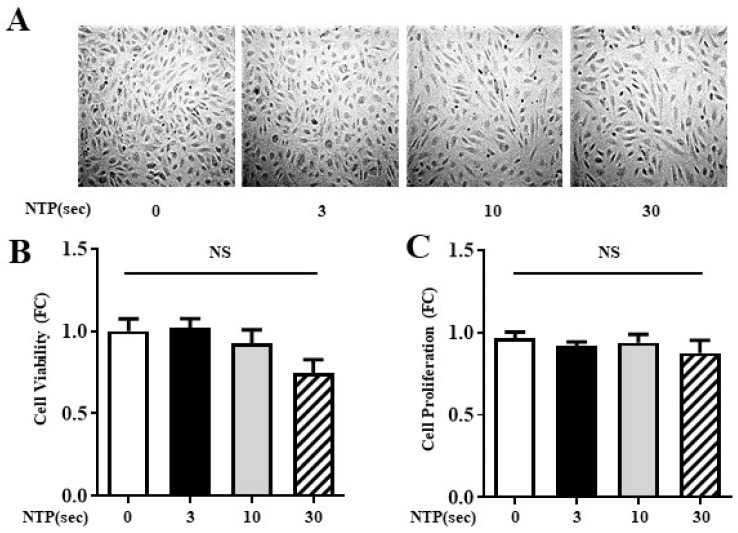
Effect of NTP on HUVEC viability and proliferation. (**A**) Representative photomicrographs of human umbilical vein endothelial cells (HUVECs) treated with NTP for 0, 3, 10, or 30 s. Images were acquired from the center of the NTP-treated spots following 12 h incubation. (**B**) Cell viability, expressed as fold change relative to untreated controls, assessed by MTT assay. (**C**) Cell proliferation, measured by CCK-8 assay. Data are presented as individual values with the mean ± SD from three independent experiments. No statistically significant differences (NS) were observed among groups.

**Figure 3 ijms-26-04449-f003:**
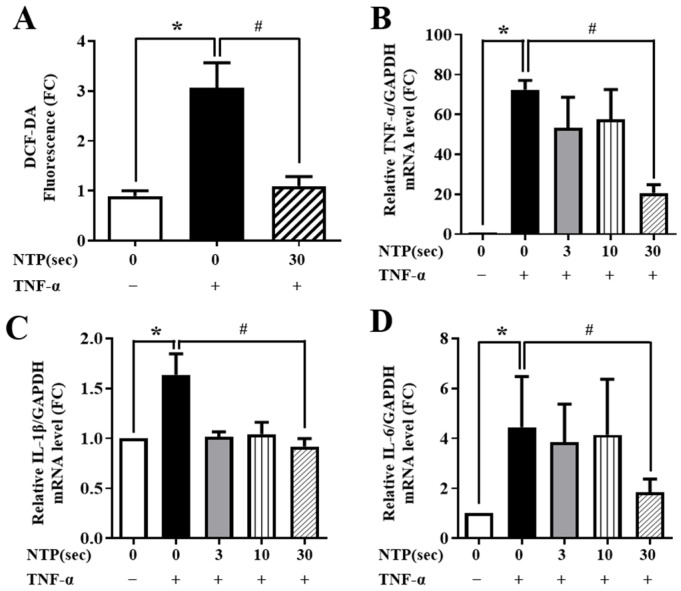
Effect of NTP on TNF-α-induced ROS generation and inflammatory cytokine expression in HUVECs. HUVECs were treated with NTP for 0, 3, 10, or 30 s, followed by TNF-α stimulation (10 ng/mL for 60 min) and 12 h incubation. (**A**) Intracellular ROS levels were measured using DCF-DA fluorescence assay. (**B**–**D**) mRNA levels of pro-inflammatory cytokines TNF-α (**B**), IL-1β (**C**), and IL-6 (**D**) were quantified by qPCR. A TNF-α-only control group (without NTP) was included. Notably, IL-1β showed significant inhibition even at 3 s, while TNF-α and IL-6 required 30 s of NTP exposure. Data are shown as individual values with the mean ± SD from three independent experiments. Statistical analysis shows that * *p* < 0.05 compared with the no-NTP treatment group, and # *p* < 0.05 compared with the TNF-α-only treatment group.

**Figure 4 ijms-26-04449-f004:**
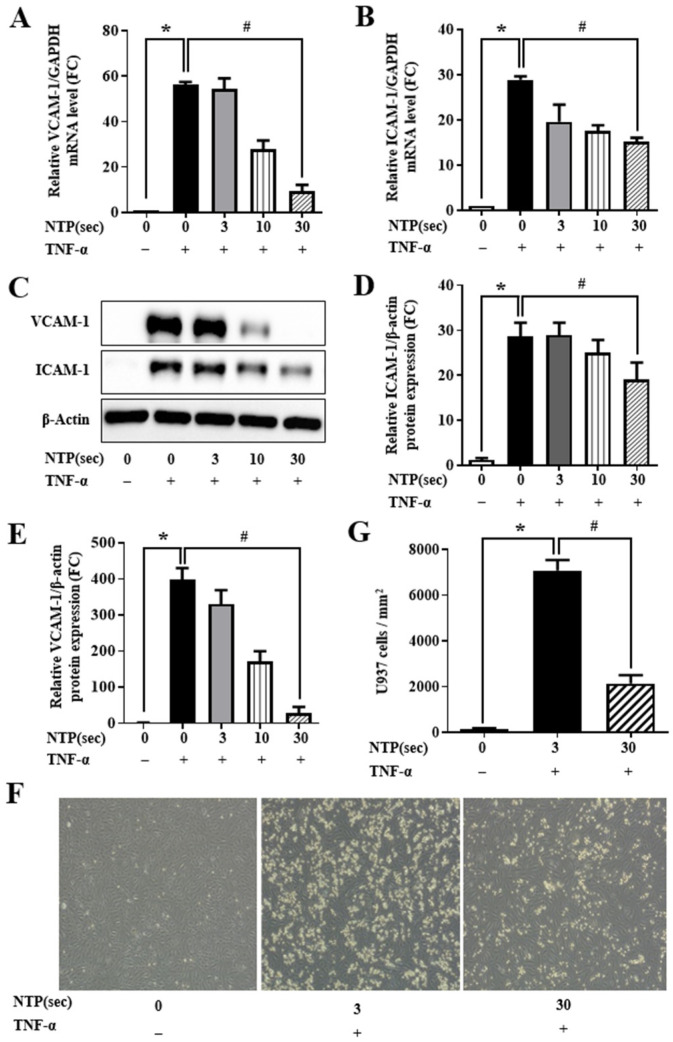
Effect of NTP on TNF-α-induced adhesion molecule expression and monocyte adhesion in HUVECs. HUVECs were treated with NTP (0, 3, 10, or 30 s), stimulated with TNF-α (10 ng/mL for 60 min), and incubated for 12 h. (**A**,**B**) qPCR analysis of mRNA levels of VCAM-1 and ICAM-1. (**C**) Western blot analysis of VCAM-1 and ICAM-1 protein expression. (**D**,**E**) Densitometric quantification using ImageJ (Quantity One 1-D software). (**F**) Representative photomicrographs showing U937 monocyte adhesion to HUVECs. (**G**) Quantification of adherent monocytes. Data are expressed as individual values with the mean ± SD from three independent experiments. Statistical analysis shows that * *p* < 0.05 compared with the no-NTP treatment group, and # *p* < 0.05 compared with the TNF-α-only treatment group.

**Figure 5 ijms-26-04449-f005:**
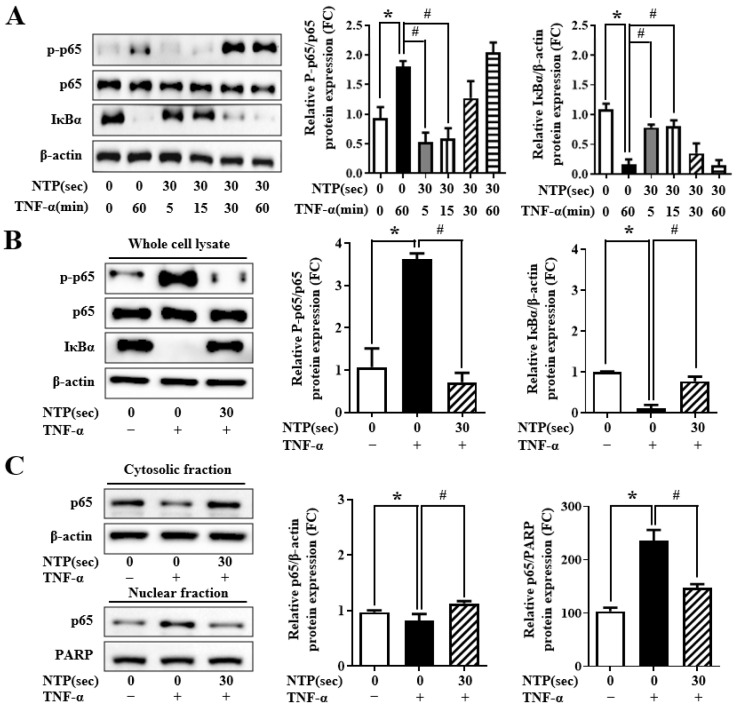
Effect of NTP on TNF-α-induced NF-κB signaling in HUVECs. HUVECs were treated with NTP for 0 or 30 s and incubated for 30 min before stimulation with TNF-α (10 ng/mL). Time course experiments were performed with TNF-α stimulation for 5, 15, 30, and 60 min, followed by protein analysis at each time point. (**A**,**B**) Western blot of phosphorylated p65 (p-p65), total p65, and IκBα in whole-cell lysates. (**C**) Western blot of p65 from cytosolic and nuclear fractions to assess nuclear translocation. (**D**) Immunofluorescence images showing localization of p65 (green) and DAPI (blue) under each treatment. Quantification of nuclear p65-positive cells is shown. Images represent three independent experiments. Statistical analysis shows that * *p* < 0.05 compared with the no-NTP treatment group, and # *p* < 0.05 compared with the TNF-α-only treatment group.

**Figure 6 ijms-26-04449-f006:**
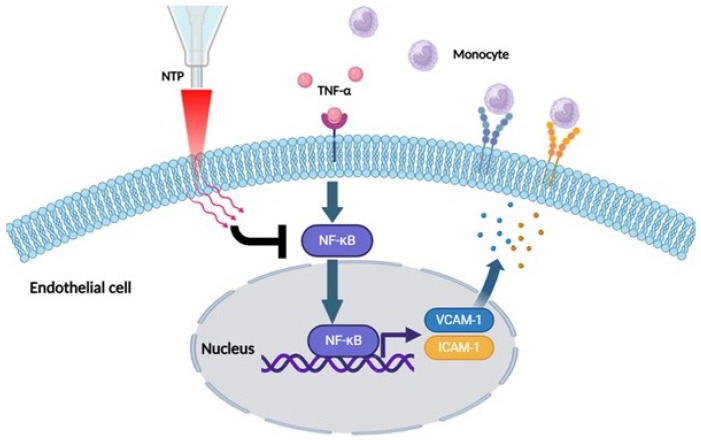
Proposed mechanism of NTP-mediated suppression of endothelial inflammation. Schematic diagram depicting the mechanism by which NTP suppresses TNF-α-induced endothelial inflammation. TNF-α activates NF-κB signaling, leading to nuclear translocation of p65 and expression of adhesion molecules (VCAM-1, ICAM-1), promoting monocyte adhesion. NTP inhibits NF-κB activation by preventing p65 translocation, thereby reducing adhesion molecule expression and inflammatory response in HUVECs.

**Table 1 ijms-26-04449-t001:** Primers used in qPCR.

Genes	Sequence	
*VCAM-1*	Sense	5′-GTTGAATGCGGGAGTAT-3′
	Antisense	5′-TTCATGTTGGCTTTTCTTGC-3′
*ICAM-1*	Sense	5′-AGAGGTTGAACCCCACAGTC-3′
	Antisense	5′-TCTGGCTTCGTCAGAATCAC-3′
*TNF-α*	Senses	5′-CCCAGGGACCTCTCTCTAATCA-3′
	Antisense	5′-AGCTGCCCCTCAGCTTGAG-3′
*IL-1β*	Sense	5′-TGGCAATGAGGATGACTTGTTC-3′
	Antisense	5′-CTGTAGTGGTCGGAGATT-3′
*IL-6*	Sense	5′-CCACTCACCTCTTCAGAACG-3′
	Antisense	5′-CATCTTTGGAAGGTTCAGGTTG-3′

## Data Availability

The datasets supporting the conclusions of this article are available from the corresponding author upon reasonable request.

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
