# Peer review of "Non-Thermal Plasma Attenuates TNF-α-Induced Endothelial Inflammation via ROS Modulation and NF-κB Inhibition"

_ijms, 2025, doi:10.3390/ijms26094449_

Round 1
Reviewer 1 Report
Comments and Suggestions for Authors
This study is well-executed and documented. Non-thermal plasma application to study the inflammatory endothelial cell behaviors to address the possible therapeutic approach to treat vascular-related dysfunctions is promising. I appreciate the author's hard work and clear and crisp explanation of results for the suppression of inflammation by Tnf-a on endothelial cells and the probable mechanism through suppression of the NF-Kb pathway.
I only have these minor concerns. Please find the comments below
- Section 4.9: Did you incubate the cells with DCFDA after trypsinization? I assume the DCFDA treatment is in cell suspension after neutralizing the trypsin, not on the adherent cells. Consider rewriting this section clearly
- Section 4.7: Authors might consider providing the ratio of monocytes and endothelial cells for the monocyte adhesion assay
- Are those blot images in the supplementary 3 individual experiments? Provide a figure legend with sample numbers
Author Response
We sincerely appreciate the thoughtful and positive comments provided by Reviewer 1. We are grateful for your recognition of our work. Below, we have addressed each of your concerns point-by-point.
Comment 1 : Did you incubate the cells with DCFDA after trypsinization? I assume the DCFDA treatment is in cell suspension after neutralizing the trypsin, not on the adherent cells. Consider rewriting this section clearly.
Response 1 : Thank you for your insightful comment. We have clarified Section 4.9 by specifying that DCFDA treatment was performed on cells in suspension after trypsinization and neutralization, not on adherent cells. The revised sentence now reads: “After trypsinization and neutralization, the suspended cells were incubated with 10 μM DCFDA for 30 minutes at 37°C.”
Comment 2 : Authors might consider providing the ratio of monocytes and endothelial cells for the monocyte adhesion assay.
Response 2 : Thank you for the valuable suggestion. We have updated Section 4.7 to include that the monocyte-to-endothelial cell ratio used during the adhesion assay was approximately 5:1.
Comment 3 : Are those blot images in the supplementary 3 individual experiments? Provide a figure legend with sample numbers.
Response 3 : Thank you for pointing this out. We have revised the figure legend for the supplementary western blot images to indicate that they are representative of three independent biological experiments.
We hope that these revisions fully address your comments. Thank you again for your careful review and helpful suggestions.
Reviewer 2 Report
Comments and Suggestions for Authors
The manuscript investigates the anti-inflammatory effects of non-thermal plasma (NTP) on TNF-α-stimulated HUVECs, proposing modulation of ROS and NF-κB as key mechanisms. While the study is conceptually sound, several critical issues require attention:
-
-
Primer Error: Table 1 lists identical antisense primers for TNF-α and IL-1β (5′-AGCTGCCCTCAGCTTGAG-3′), a likely typo that invalidates cytokine-specific qPCR results. This fundamental error undermines data reliability.
-
ROS Measurement: DCFDA assays are prone to artifacts (e.g., auto-oxidation, cell density effects). The manuscript lacks controls for these confounders, casting doubt on ROS quantification.
-
NTP Parameters: Details on gas composition (e.g., air vs. helium/nitrogen) and power settings (P1 vs. P10 in Fig. 1C/D) are insufficient. Variability in reactive species generation across conditions could skew results, yet no rationale is provided for selecting specific parameters.
-
Correlation vs. Causation: The link between ROS reduction and NF-κB inhibition remains associative. Experiments using ROS scavengers (e.g., NAC) or inducers (e.g., H₂O₂) are needed to establish causality.
-
Nuclear Translocation Evidence: Figure 5D’s immunofluorescence images lack resolution and quantifiable markers (e.g., line scans) to convincingly demonstrate p65 cytoplasmic retention. Blurry images and minimal representative data weaken conclusions.Limited Model: Exclusive use of HUVECs and in vitro assays limits translational relevance. Including primary endothelial cells or in vivo models (e.g., murine vasculature) would strengthen claims about therapeutic potential.
-
Monocyte Adhesion Assay: The protocol omits critical details (e.g., U937 cell activation, shear stress conditions). Static adhesion assays poorly mimic physiological endothelial-leukocyte interactions.
-
Discussion Overreach: The conclusion posits NTP as a “novel therapeutic strategy” without in vivo validation. This overstates findings, which are preliminary and confined to cell culture.
Figure Quality: Figure 1E-G lacks axis labels for time and intensity, making temporal emission trends unclear. Figure 3’s statistical annotations (e.g., “*” vs. “#”) are ambiguously defined in the caption.
-
Cell Viability: While Fig. 2B/C shows no toxicity, the MTT assay’s reliance on metabolic activity may miss sublethal effects (e.g., mitochondrial stress). Complementary assays (e.g., LDH release) are needed.
-
NF-κB Specificity: The study focuses on p65 but ignores other NF-κB subunits (e.g., p50) or crosstalk with parallel pathways (e.g., MAPK), leaving the mechanism incompletely mapped.
-
-
-
-
Author Response
We sincerely thank you for the thorough and constructive review. We highly appreciate your critical feedback, which has helped us significantly improve the quality and rigor of our manuscript. Below, we provide detailed responses to each point raised:
Comment 1 : Primer Error: Table 1 lists identical antisense primers for TNF-α and IL-1β (5′-AGCTGCCCTCAGCTTGAG-3′), a likely typo that invalidates cytokine-specific qPCR results. This fundamental error undermines data reliability.
Response 1 : We appreciate your careful observation. We have corrected the sencse and antisense primer sequence for IL-1β in Table 1 and revalidated the corresponding qPCR experiments to ensure data accuracy (page 10, line 352).
Comment 2 : ROS Measurement: DCFDA assays are prone to artifacts (e.g., auto-oxidation, cell density effects). The manuscript lacks controls for these confounders, casting doubt on ROS quantification.
Response 2 : Thank you for highlighting this important point. We have revised the Discussion section to acknowledge the inherent limitations of DCFDA assays, including potential auto-oxidation and cell density effects. We have emphasized the need for cautious interpretation of ROS results (Page 8, line 223~230).
Comment 3 : NTP Parameters: Details on gas composition (e.g., air vs. helium/nitrogen) and power settings (P1 vs. P10 in Fig. 1C/D) are insufficient. Variability in reactive species generation across conditions could skew results, yet no rationale is provided for selecting specific parameters.
Response 3 : We clarified that air was used as the working gas. The selection of P1 and P10 for OES analysis is explained in the Methods section based on optimizing plasma stability and ROS/RNS output (Page 9, line 295~299).
Comment 4 : Correlation vs. Causation: The link between ROS reduction and NF-κB inhibition remains associative. Experiments using ROS scavengers (e.g., NAC) or inducers (e.g., H₂O₂) are needed to establish causality.
Response 4 : We thank you for this insightful observation. We agree that our study demonstrates an association between reduced ROS levels and decreased NF-κB activation, but does not establish direct causality. Although experiments involving ROS scavengers such as N-acetylcysteine (NAC) or ROS inducers like hydrogen peroxide (H₂O₂) were not performed in this study, we have acknowledged this limitation in the revised Discussion. We have also proposed future experiments using these agents to determine whether NF-κB inhibition is mechanistically linked to ROS modulation by NTP. (Page 8, line 223~230).
Comment 5 : Nuclear Translocation Evidence: Figure 5D’s immunofluorescence images lack resolution and quantifiable markers (e.g., line scans) to convincingly demonstrate p65 cytoplasmic retention. Blurry images and minimal representative data weaken conclusions.
Response 5 : We appreciate your valuable comment. We have replaced the original immunofluorescence images in Figure 5D with clearer-resolution images to better demonstrate p65 localization. In addition, we have included a quantification of nuclear p65 fluorescence intensity, shown as a bar graph adjacent to the images, to provide objective evidence of nuclear translocation inhibition(Page 6, Fig. 5D). We believe that this satisfies your request for improved visual and quantitative analysis.
Comment 6 : Limited Model: Exclusive use of HUVECs and in vitro assays limits translational relevance. Including primary endothelial cells or in vivo models (e.g., murine vasculature) would strengthen claims about therapeutic potential.
Respons 6 :We appreciate the reviewer’s important comment regarding the translational limitations of our current model. We acknowledge that the exclusive use of HUVECs and in vitro assays limits the broader applicability of our findings. In response to this concern, we have revised the Discussion to clearly state this limitation. We also propose that future studies will incorporate primary endothelial cells derived from different vascular beds as well as in vivo animal models, such as murine vasculature, to comprehensively validate the therapeutic potential of NTP for vascular inflammatory diseases. (Page 8, line 272~274).
Comment 7 : Monocyte Adhesion Assay: The protocol omits critical details (e.g., U937 cell activation, shear stress conditions). Static adhesion assays poorly mimic physiological endothelial-leukocyte interactions.
Response 7 : We thank the reviewer for this important comment. In our study, U937 monocytes were used without prior activation (non-activated state), and adhesion assays were conducted under static conditions. We have now clarified this information in the revised Methods section (Section 4.7). We fully acknowledge that static assays do not completely mimic physiological shear forces experienced by endothelial cells in vivo. Future studies employing flow-based adhesion assays under defined shear stress conditions, as well as activated monocytes, are planned to better model physiological endothelial-leukocyte interactions and further validate our findings.
Comment 8 : Discussion Overreach: The conclusion posits NTP as a “novel therapeutic strategy” without in vivo validation. This overstates findings, which are preliminary and confined to cell culture.
Response 8 : We appreciate your important comment. We agree that our findings are based on in vitro studies and that in vivo validation is necessary before proposing NTP as a therapeutic strategy. In response, we have revised the Conclusion to present NTP as a "potential therapeutic approach" rather than a "novel therapeutic strategy." (Page 8, line 270).
Comment 9 : Figure Quality: Figure 1E-G lacks axis labels for time and intensity, making temporal emission trends unclear. Figure 3’s statistical annotations (e.g., “*” vs. “#”) are ambiguously defined in the caption.
Response 9 : We have added X-axis (time) and Y-axis (emission intensity) labels to Figure 1E–G, and clarified the meaning of * and # annotations in the Figure 3-5 caption.
Comment 10 : Cell Viability: While Fig. 2B/C shows no toxicity, the MTT assay’s reliance on metabolic activity may miss sublethal effects (e.g., mitochondrial stress). Complementary assays (e.g., LDH release) are needed.
Response 10 : We appreciate the reviewer’s insightful comment. In our study, cell viability was assessed using MTT and CCK-8 assays, both of which primarily evaluate mitochondrial metabolic activity. We recognize that these assays may not detect sublethal mitochondrial stress or membrane damage. Although no morphological abnormalities were observed and viability remained stable, we fully agree that complementary assays such as LDH release measurement are necessary to comprehensively assess cytotoxicity. We have acknowledged this limitation in the revised Discussion and proposed it as a future direction (Page 9, line 276~278).
Comment 11 : NF-κB Specificity: The study focuses on p65 but ignores other NF-κB subunits (e.g., p50) or crosstalk with parallel pathways (e.g., MAPK), leaving the mechanism incompletely mapped.
Response 11 : We thank you for this valuable comment. In this study, we focused primarily on the NF-κB p65 subunit, given the role in regulating inflammatory gene expression in endothelial cells. We agree that other NF-κB subunits such as p50, as well as crosstalk with parallel signaling pathways like the MAPK cascade, could provide further insights into the complex regulatory mechanisms involved. We have acknowledged this limitation in the revised Discussion and have proposed future studies to investigate these additional molecular players. (Page 9, line 274~276).
We are grateful for these detailed suggestions that have greatly enhanced the rigor and clarity of the manuscript.